# The Biosynthesis Process of Small RNA and Its Pivotal Roles in Plant Development

**DOI:** 10.3390/ijms25147680

**Published:** 2024-07-12

**Authors:** Quan Li, Yanan Wang, Zhihui Sun, Haiyang Li, Huan Liu

**Affiliations:** Guangdong Key Laboratory of Plant Adaptation and Molecular Design, Guangzhou Key Laboratory of Crop Gene Editing, Innovative Center of Molecular Genetics and Evolution, School of Life Sciences, Guangzhou University, Guangzhou 510006, China; liquan@gzhu.edu.cn (Q.L.); a137810907@163.com (Y.W.); szh@gzhu.edu.cn (Z.S.)

**Keywords:** biosynthesis, function, miRNA, plant development, siRNA

## Abstract

In the realm of plant biology, small RNAs (sRNAs) are imperative in the orchestration of gene expression, playing pivotal roles across a spectrum of developmental sequences and responses to environmental stressors. The biosynthetic cascade of sRNAs is characterized by an elaborate network of enzymatic pathways that meticulously process double-stranded RNA (dsRNA) precursors into sRNA molecules, typically 20 to 30 nucleotides in length. These sRNAs, chiefly microRNAs (miRNAs) and small interfering RNAs (siRNAs), are integral in guiding the RNA-induced silencing complex (RISC) to selectively target messenger RNAs (mRNAs) for post-transcriptional modulation. This regulation is achieved either through the targeted cleavage or the suppression of translational efficiency of the mRNAs. In plant development, sRNAs are integral to the modulation of key pathways that govern growth patterns, organ differentiation, and developmental timing. The biogenesis of sRNA itself is a fine-tuned process, beginning with transcription and proceeding through a series of processing steps involving Dicer-like enzymes and RNA-binding proteins. Recent advances in the field have illuminated the complex processes underlying the generation and function of small RNAs (sRNAs), including the identification of new sRNA categories and the clarification of their involvement in the intercommunication among diverse regulatory pathways. This review endeavors to evaluate the contemporary comprehension of sRNA biosynthesis and to underscore the pivotal role these molecules play in directing the intricate performance of plant developmental processes.

## 1. Introduction

sRNAs are found across all domains of life, including in bacteria, archaea, and various eukaryotes. Their diverse compositions and functions have continued to astonish researchers over the past two decades [1]. Synthesized by the RNAIII enzyme, these sRNAs are excised from dsRNA or single-stranded RNA (ssRNA) harboring a stem–loop architecture [2,3]. As non-coding entities, sRNAs pervade plant cellular machinery, orchestrating the regulation of target gene mRNAs—a process central to gene expression control, genomic integrity, stress response, and a myriad of vital biological functions [4,5,6]. The unearthing of a vast spectrum of sRNAs in the plant has elucidated the complexities inherent in plant gene regulatory mechanisms. This discovery has significantly propelled the field of agricultural biotechnology forward, offering novel insights and applications in the modulation and enhancement of plant traits [7,8,9].

Amidst the vast constellation of non-coding RNAs, this review narrows its focus to plant sRNAs, elucidating their biosynthetic origins and biological mandates, specifically miRNAs and siRNAs [6]. The sRNA processing imparts distinct length characteristics to these molecules. Contrastingly, the preferential selection of the initial nucleotide at the sRNA’s 5′-terminus is pivotal in dictating the recruitment of Argonaute (AGO) proteins, thereby delineating the sRNA’s functional spectrum [7,8]. miRNAs are single-stranded, non-coding RNA molecules containing 20 to 24 nucleotides. They are widely found in eukaryotes and play a crucial role in regulating gene expression. Derived from RNA transcripts, miRNAs bind specifically to target mRNA, inhibiting post-transcriptional gene expression. Their functions extend to cell cycle regulation and developmental timing in organisms [9]. miRNAs are engendered through the precise processing of stem–loop structured precursor RNAs (pre-miRNAs), which are transcribed by the esteemed RNA polymerase II [9,10]. siRNA is a class of double-stranded RNA molecule, typically 21–24 base pairs long. Similar to miRNA, siRNA functions within the RNA interference (RNAi) pathway. It specifically targets and degrades mRNAs that express genes with complementary nucleotide sequences, thereby inhibiting translation [10]. Concurrently, the origin of siRNAs is ascribed to the cleavage of dsRNA substrates, underscoring the divergent pathways that shape the sRNA landscape [6,10]. To enhance our understanding of sRNAs and facilitate their application in agricultural production, we offer an overview of sRNA biogenesis, transport processes, and the underlying mechanisms of their functions.

## 2. sRNA

### 2.1. miRNA

miRNAs, a cadre of diminutive RNA entities, span approximately 20–24 nucleotides. In the verdant realm of plants, these miRNAs typically manifest at a length of 21 nucleotides and are pivotal in the orchestration of gene regulatory networks. Through their affinity for target gene mRNAs, miRNAs enact a process of RNA-induced silencing [11]. The molecular dialogue between miRNAs and their cognate target sequences is governed by the extent of nucleotide complementarity, which consequently determines the fate of both the miRNA and the target mRNA. In plants, high-precision alignments between miRNAs and their targets instigate the total disintegration of the target gene’s transcript. Conversely, miRNAs that exhibit partial complementarity to their targets predominantly attenuate the translational process of the corresponding gene products (Figure 1) [10,12].

Within the elaborate choreography of gene expression, miRNA genes are initially transcribed by RNA polymerase II into primary transcripts, known as pri-miRNAs, which consist of several hundred nucleotides. These nascent transcripts are then refined by the RNaseIII family’s Dicer-like enzymes, predominantly DCL1, into pre-miRNA. Subsequently, DCL1 meticulously carves the pre-miRNA into a double-stranded configuration, which is then elegantly modified by the addition of a methyl group at the 3′-terminus’s ultimate nucleotide, a reaction catalyzed by the miRNA methyltransferase HUA ENHANCER 1 (HEN1) [13,14]. In the nucleus, mature miRNA strands are integrated into ARGONAUTE 1 (AGO1). Within the nucleus, these mature miRNA strands are assimilated into ARGONAUTE 1 (AGO1), completing their journey to functionality. In this final act of the post-transcriptional gene regulation ballet, the mature miRNA, ensconced within the ARGONAUTE 1 (AGO1) complex, executes its role with precision. It either catalyzes the mRNA excision—a process akin to molecular surgery, excising the target mRNA with exactitude—or it imposes translational repression, subtly silencing the mRNA without cleavage. Through these mechanisms, miRNAs exert their influence, conducting the intricate symphony of gene silencing that fine-tunes the expression of the plant’s genetic repertoire [15,16,17,18].

Currently, it is acknowledged that a multitude of miRNAs with well-characterized functions exhibit a high degree of conservation; however, this is not a universal trait among plant miRNAs. While many miRNAs are conserved, a significant number are not [19]. A considerable proportion of plant miRNAs are endemic, flourishing solely within their native species and a select cadre of closely allied taxa. These miRNAs, termed species-specific miRNAs, often elude identification of their target genes or may not be associated with any targets at all. Moreover, the expression levels of these species-specific miRNAs are markedly subdued when contrasted with their conserved counterparts [20,21].

### 2.2. siRNA

In plants, siRNAs are critical modulators of gene silencing, comprising 21 to 24 nucleotides derived from dsRNA precursors. Predominantly, these endogenous siRNAs originate from genomic repetitive sequences and transposable element (TE) intergenic regions, with a characteristic length of 24 nucleotides. The biogenesis of these molecules involves their generation from dsRNA and subsequent processing via Dicer-like 3 (DCL3) enzyme cleavage. Intriguingly, the majority of 24-nucleotide miRNAs identified in both *Arabidopsis* and rice exhibit species-specific patterns. This specificity implies that these 24-nucleotide miRNAs may represent evolutionary novelties, having emerged recently and, thus, not having been subjected to extensive evolutionary selection, as suggested by recent studies (Figure 1) [22,23].

The synthesis of plant siRNAs is distinct from that of miRNAs, primarily because siRNAs are typically derived from long non-coding RNAs (lncRNA), regions proximal to heterochromatin, repetitive sequences, and TEs within the genome [24,25]. In the intricate regulatory networks of plant genomics, the plant-specific RNA polymerase IV (Pol IV) is of paramount importance. It is strategically recruited to the loci designated for RNA-directed DNA methylation (RdDM), where it initiates the synthesis of single-stranded siRNA precursors. This recruitment is not merely a procedural formality but a critical juncture in the pathway, pivotal for the subsequent steps that culminate in the nuanced orchestration of gene expression regulation [26]. In the context of plant molecular biology, the transformation of single-stranded siRNA precursors into dsRNA is facilitated by RNA-dependent RNA polymerase 2 (RDR2). Following this, the DCL3 enzyme meticulously processes the dsRNAs, culminating in the production of 24-nucleotide mature siRNAs [27,28]. These siRNA molecules are subsequently methylated by HEN1 and then translocated to the cytoplasm [29]. Upon integration with the RISC, the dsRNA is bifurcated into single strands. One strand assumes the role of the guide strand within the RISC, while its counterpart is relegated to the passenger strand [30]. Concurrently, the plant-specific Pol V engages in the transcription of non-coding RNAs at the RdDM loci. These non-coding RNAs are pivotal in magnetizing the siRNA–AGO4 complex, a process orchestrated by the nucleotide complementarity between the non-coding RNAs and the guide strand of the RISC. This molecular affinity facilitates the enlistment of DNA Remethylase 2 (DRM2), thereby initiating the process of DNA methylation at the RdDM sites [31].

Phased small interfering RNAs (phasiRNAs) represent a distinct subset of endogenous siRNAs that exert a pivotal regulatory influence in plants. A subset of phasiRNAs has been documented to orchestrate the degradation of complementary mRNAs, a critical aspect of gene regulation [32]. The genesis of phasiRNAs is predicated upon the meticulous cleavage of either target mRNA or lncRNA by a 22-nucleotide miRNA. This cleavage event triggers the conversion of the cleaved RNA into dsRNA through the enzymatic activity of RDR6. The dsRNA is subsequently subjected to systematic cleavage by Dicer-like proteins, notably DCL4, culminating in the formation of a double-stranded phasiRNA body. Analogous to the miRNA double-stranded bodies, the RISC laden with phasiRNA engages in homologous base pairing with the target RNA, leading to its degradation [32].

LncRNAs are associated with sRNAs due to their roles as sRNA targets, sRNA precursors, or miRNA sponges. These interactions contribute to the regulation of plant immunity and growth. Experimental evidence has revealed specific connections between lncRNAs and sRNAs in plants [33]. In addition to miRNAs, lncRNAs can also be targeted by siRNAs. Notably, vsiRNA-mediated lncRNA cleavage seems to play an emerging regulatory role for siRNAs. Previous studies have demonstrated that virus-infected plants exhibit an accumulation of viral siRNAs. These viral siRNAs, which exhibit high sequence complementarity to host genes or host gene promoters, can induce the cleavage of host mRNAs or transcriptional inactivation of host gene promoters [33].

## 3. Key Protein Involved in the Biosynthesis of sRNAs

In the realm of plant biology, a diverse array of sRNAs are generated via distinct biogenesis pathways (Figure 2). A cadre of proteins, each playing an indispensable role in these pathways, have been identified. Within the intricate molecular machinery of plant cells, RNA polymerases and RNA-dependent RNA polymerases (RDRPs) are indispensable catalysts in the transcriptional genesis of sRNA precursors. These enzymes are central to the complex network of pathways that facilitate the synthesis and subsequent processing of sRNAs, thereby underpinning the sophisticated regulatory systems that modulate gene expression through sRNA-mediated interactions. Dicer/Dicer-like proteins, belonging to the ribonuclease class III, are responsible for the cleavage of these sRNA precursors. The methylesterase HEN1 contributes to the stabilization of sRNA, while the ARGONAUTE (AGO) protein is pivotal in the formation of the RISC with sRNA [34].

### 3.1. RNA Polymerases (Pols)

In the botanical realm, plants possess a quintet of RNA polymerases, each with distinct roles: Pol I–III, which are ubiquitous among eukaryotic organisms, and Pol IV and V, which are unique to the plant kingdom. Specifically, Pol I is tasked with the transcription of the 45S rRNA genes, a critical component of protein synthesis machinery. In contrast, Pol II oversees the transcription of the majority of genes, including those encoding proteins. Pol III transcribes shorter structural RNAs, such as tRNA and 5S rRNA [35,36]. Within the complex tapestry of plant genetic regulation, a cadre of RNA polymerases meticulously orchestrates the synthesis of small RNA (sRNA) precursor sequences. Pol II is primarily responsible for the transcription of microRNA (miRNA) precursors. Conversely, the transcriptional duties for small interfering RNA (siRNA) precursors are allocated to either Pol IV or Pol V [24,31]. Both Pol IV and V are critical to the RdDM pathway; Pol IV is indispensable for the biogenesis of siRNAs, while Pol V synthesizes non-coding transcripts that act as scaffolds for the genesis of 24-nucleotide siRNAs [31]. This intricate interplay of RNA polymerases underscores the complexity of sRNA biogenesis in plants.

The eukaryotic Pol I–III protein complex is composed of 12–17 proteins [37,38]. In *Arabidopsis*, the principal subunits of RNA polymerases I–III—specifically NRPA1, NRPB1, and NRPC1—exhibit evolutionary conservation with the β’ subunit of prokaryotic RNA polymerase. The subunits of secondary magnitude, namely NRPA2, NRPB2, and NRPC2, display homology with the β subunit of the same enzyme [39]. The Pol I–III complex is indispensable for the transcription of life-sustaining genes. Mutations in the secondary largest subunits of RNA polymerases I–III in *Arabidopsis*, namely *nrpa2*, *nrpb2*, and *nrpc2*, all result in lethality [40]. In the specialized ensemble of plant RNA polymerases, Pol IV and V are distinguished by their composition of 12 subunits, which exhibit homology with those of Pol II. Pol IV is characterized by four unique subunits that differentiate it from Pol II, whereas Pol V is defined by six distinct subunits. In contrast to Pol I–III, mutations in the largest and second-largest subunits of Pol IV and V do not result in lethality, indicating their non-essential nature for plant viability. Moreover, the functional roles of Pol IV and V are distinct and do not intersect with the fundamental processes governed by Pol I–III [41]. In the *nrpd1* mutant of *Arabidopsis*, the expression level of 24 nt sRNA is significantly reduced, and the level of CHG methylation is decreased [42]. Transcription products generated by Pol IV are shorter in length (mostly 30–40 nt) compared to those produced by Pol II, suggesting that a single transcription product from Pol IV can yield only one mature siRNA [43]. In the context of *Arabidopsis thaliana*, mutants of RNA Pol IV, specifically *nrpd1* and *nrpd2a*, do not manifest substantial phenotypic growth disparities when juxtaposed with the wild-type counterparts. However, in maize, the Pol IV mutant *rmr6* exhibits discernible developmental aberrations at the internodes. Similarly, the tomato *slnrpd1* mutant is characterized by marked dwarfism, highlighting the divergent phenotypic consequences of Pol IV mutations across different plant species [44,45]. Pol V, despite being less studied, has been implicated in the regulation of gene expression regulation and heterochromatin formation in a Pol IV-independent manner, although the exact mechanism remains elusive [46].

### 3.2. RNA-Dependent RNA Polymerase (RdRP)

RdRP is a distinct class of RNA polymerase protein and is found in organisms possessing a conserved catalytic domain capable of synthesizing complementary dsRNA molecules from ssRNA templates [47]. RdRP primarily orchestrates the synthesis of siRNA in plants. In the presence of RdRP, ssRNA and primary siRNA molecules can engender copious amounts of dsRNA, which are subsequently processed by Dicer-like proteins to yield siRNAs [22]. siRNAs can be amplified by RdRP, which uses mRNA as a template and siRNA as a primer to generate sufficient dsRNA for Dicer to produce additional siRNAs. These siRNAs can then form RISCs and continue to degrade mRNA, thereby amplifying the siRNA-mediated silencing cascade [48]. In *Arabidopsis*, the RdRP family encompasses six distinct members, designated as *RDR1* through *RDR6*. *RDR1* is principally involved in the amplification of exogenous siRNAs, playing a pivotal role in the plant’s defense mechanisms against viral pathogens. On the other hand, *RDR2* is implicated in the process of transcriptional gene silencing (TGS), a critical epigenetic regulatory mechanism that modulates gene expression at the transcriptional level. The functional dichotomy between RDR1 and RDR2 highlights the specialized roles these enzymes play within the broader context of RNA-mediated gene regulation in plants. RDR6 participates in the biosynthesis of phased, secondary phasiRNAs in *Arabidopsis*. To date, no functions have been identified for *Arabidopsis* RDR3, RDR4, and RDR5 in the gene silencing pathway [49,50,51].

In eukaryotic organisms, *RDRPs* are classified into three primary subfamilies: *RDRα*, *RDRβ*, and *RDRγ*. Plants typically harbor multiple *RDRs*. The *RDRα* subfamily, which includes *RDR1/2/6*, is characterized by a conserved C-terminal catalytic motif, DLDGD. The *RDRγ* subfamily, comprising *RDR3/4/5*, possesses an atypical catalytic motif, DFDGD [52]. Despite sharing the same catalytic motif, the functions of RDR1/2/6 are somewhat divergent. They participate in distinct endogenous silencing pathways in conjunction with specific Dicer-like proteins. RDR1/2/6 contribute to plant resistance against viruses and fungi. Mutations in RDR1/2/6 diminish plant resistance to DNA and RNA viruses [53]. Members of the RDRα family can indirectly influence plant resistance to fungi, bacteria, and nematodes by modulating endogenous sRNA levels [54]. In biotic stresses, the RDRα family impact abiotic stress responses in plants. Various abiotic stresses and hormones induce the expression level of *Arabidopsis RDR1*. Rice *RDR1* mutant exhibit increased sensitivity to salt and enhanced tolerance to heavy metal ions, such as copper and mercury [55]. Rice RDR6 mutants display spikelet abnormalities at high temperatures [56]. The rdr1 rdr2 rdr6 triple mutant of *Arabidopsis* exhibits significantly inhibited root length under polyethylene glycol (PEG) treatment [57]. While numerous studies have found that the RDRα family can influence sRNA synthesis, mutants of *Arabidopsis* and rice *RDRα* members did not exhibit severe development defects. However, maize *RDR6* mutants displayed pronounced developmental abnormalities [58,59]. Phylogenetic scrutiny has uncovered variations in the number of *RDR 1/2/6* among different species [60]. The operational dynamics of the *RDRγ* subfamily are yet to be fully deciphered, accentuating the imperative for augmented investigative efforts to demystify the contributions of the RDR contingent to the intricate regulatory networks within plants.

### 3.3. Dicer/Dicer-like (DCLs)

Dicer-like proteins, essential to the biosynthesis of sRNAs, are nucleic acid endonucleases that belong to the RNaseIII family. They exhibit specificity in recognizing and processing dsRNA, initiating sRNA production through the cleavage of dsRNA precursors [61]. In the verdant realm of plant biology, a minimum of four gene family members are known to encode the Dicer-like proteins. DCL1 is tasked with the generation of 21-nucleotide miRNAs, which are processed from primary transcripts exhibiting stem–loop configurations. DCL2 is responsible for the production of 22-nucleotide siRNAs, playing an integral role in certain facets, such as plant growth, development, antiviral defense, and stress resilience. DCL3 is involved in the synthesis of 24-nucleotide siRNAs, which are key components in RdDM. DCL4 produces 21 nt siRNAs that guide mRNA cleavage and also contribute to antiviral defense [61]. Interestingly, certain plant species encode a greater diversity of Dicer-like proteins; for example, the soybean genome harbors seven distinct DICER-LIKE protein genes [62].

A prototypical Dicer protein is distinguished by its composite structural domains, which include a decapping enzyme domain, a domain of unknown function (DUF283), a PAZ domain, dual RNase III domains, and a domain for double-stranded RNA binding [63]. These domains equip DCL with the ability to accurately recognize and process dsRNA substrates into sRNA. The PAZ and RNase III domains are integral to the dsRNA cleavage process. The PAZ domain specifically recognizes and binds the 2-nucleotide overhang at the 3′ termini of the dsRNA precursor. Concurrently, the bifunctional RNase III domains orchestrate the cleavage of both strands of the dsRNA. This conformation acts as a molecular caliper, dictating the dimension of the resultant sRNA product. Structural analysis has revealed additional dsRBDs in the C-terminal region [64,65]. Expression of *DCL* in plant tissues is contingent upon the developmental stage and stress response. DCL3, the enzyme responsible for generating the most copious siRNAs in plants, occupies a critical position in TGS via RdDM pathway. Upon encountering virus infestation, DCL2 and DCL4 generate copious amounts of virus-derived siRNAs to mount a defense [66]. Recent scholarly investigations have elucidated that DCL2 exerts a regulatory effect on the biosynthesis of 22-nucleotide siRNAs, which are instrumental in targeting and impeding the translation of the genes *NIA1* and *NIA2*. This ultimately curtails the efficiency of nitrogen assimilation, ensuring plant survival. Furthermore, mutations in the soybean genes *GmDCL2a* and *GmDCL2b* precipitate the absence of 22-nucleotide small interfering RNAs (siRNAs), culminating in a marked augmentation of ketone synthase within the seed coat. This biochemical change is concomitant with a phenotypic alteration, transmuting the seed coat color from yellow to brown [62]. These discoveries shed new light on the role of DCL proteins in plant biology.

### 3.4. HUA ENHANCER1 (HEN1)

HEN1, the inaugural sRNA methyltransferase identified in *Arabidopsis*, adds a methyl group to the 2′-OH at the 3′-termini of each strand of Dicer-like-cleaved sRNA precursors. This methylation prevents labelling by HESO1 and subsequent degradation by sRNA Degrading Nuclease1 (SDN1) [67]. Mutations in HEN1 lead to miRNA instability and heterogeneity at the 3′-termini. Sequencing analysis reveals a significant addition of uracil, a process known as uridylation, catalyzed by HESO1, the primary terminal uridylyltransferase in *Arabidopsis* [68]. The catalytic locus of HEN1 is uniquely differentiated from the ‘KDK’ motif that characterizes the active site within the RFM superfamily of 2′-O-methyltransferases (2′-O-MTases), as well as from the established active sites of 2′-O-MTases belonging to the SPOUT superfamily [69].

*Arabidopsis* HEN1, a pivotal player in sRNA biosynthesis, exhibits a quintet of structural domains. Among these, four domains engage directly with sRNA duplexes: a pair of double-stranded RNA-binding domains (dsRBDs), a conserved methyltransferase (MTase) domain, and a La-motif-containing domain (LCD) that discerns the 3′ terminal hydroxyl (-OH) group. The quinary domain, a peptidyl-prolyl isomerase (PPIase)-like domain (PLD), remains unaffiliated with dsRNA binding [69]. In vitro co-crystallization analyses have demonstrated that HEN1 accommodates dsRNA substrates in a monomeric form. The dsRBDs annex to both flanks of the A-form sRNA substrate, while the LCD domain sheathes the 3′ termini of the non-methylated strand. The MTase domain’s active site encases the protruding bi-nucleotide terminus of the methylated strand [70]. The determination of substrate length by HEN1 is potentially contingent upon the spatial interplay between the MTase and LCD domains, in addition to the cap site [30]. Methylation modification by HEN1 is indeed pivotal in sRNA biosynthesis and plant function. However, the mechanism of factor recruitment remains an open question. *HEN1* expression is regulated by light, and it finetunes photomorphogenesis by regulating *miR157d* and *miR319* expression [71,72]. Deletion mutants of *HEN1* manifest a spectrum of developmental anomalies, including aberrant photomorphogenesis, diminutive plant stature, delayed flowering, and infertility [25]. Conversely, deletion mutants of *HEN1 SUPPRESSOR 1* (*HESO1*) mitigate the phenotypic manifestations of *HEN1*. Overexpression of *HESO1* within *HEN1* mutants exacerbates the reduction in miRNA abundance and intensifies the developmental aberrations, thereby implicating uridylation as a precipitant of miRNA degradation [73]. This intricate interplay of molecular interactions underscores the complexity and sophistication of sRNA biogenesis in plants.

### 3.5. Argonaute (AGO)

*Arabidopsis* is endowed with ten AGO proteins, which are associated with RNA silencing and form RISCs [74]. Argonaute (AGO) proteins are categorized into three distinct clades: AGO1, 5, 8, and 10; AGO2, 3, and 7; and AGO4, 6, 8, and 9 [10]. *AGO1*, the preeminent member of the *AGO* family, orchestrates the assembly of miRNAs into RISCs and exerts regulatory control over gene expression at the post-transcriptional and translational echelons [75]. *AGO10* positively regulates the expression levels of HD-ZIP-III-like transcription factors and mediates the translational repression of several genes, including *AGO1* [76]. *AGO5*, expressed during macro sporogenesis, may be involved in gametophyte development or cell differentiation [77]. AGO2, which loads miRNAs with an A at the 5′-termini, plays a role in the plant immune system [78]. It can bind virally produced siRNAs (vsiRNAs) and dsDNA break-induced sRNAs (diRNAs) [79]. AGO3, which shares a highly similar protein sequence with AGO2, may have redundant functions [80]. AGO7 loads trans-acting siRNAs (ta-siRNAs) generated by TAS3 [81]. AGO4, 6, and 9 predominantly load 24 nt siRNAs with an A at the 5′-termini, processed by DCL3 [82]. AGO4 exerts a substantial effect on RdDM and the TGS of transposable elements [83]. Beyond its role in siRNA incorporation, AGO4 is capable of accommodating a minor subset of 24-nucleotide miRNAs and effectuating sequence-specific DNA methylation [84]. It can also replace AGO1 and AGO7 in the biosynthesis of trans-acting siRNAs (ta-siRNAs) produced by *miR172* and *miR390* [85]. High-throughput sequencing data analysis revealed that AGO4/6/9 loaded 24 nt siRNAs from different production sites [82]. Using the AGO4 promoter to initiate the expression of *AGO6* and *AGO9* reduced this difference, suggesting that *AGO4/6/9* mediate the RdDM pathway in specific tissues [86]. *AGO8*, which exhibits low expression levels in all tissues and periods in *Arabidopsis*, is generally considered to be a pseudogene with no reported function [87].

The AGO protein, a cornerstone of RNA silencing, is characterized by four major structural domains: an adaptable N-terminal domain, a conserved PAZ domain, a conserved MID domain, and a conserved PIWI domain. The conformation of the folded AGO protein assumes a bilobate architecture, engendering a channel conducive to RNA engagement [88]. The MID domain is responsible for the recognition of the 5′ termini of the small RNA (sRNA), whereas the PAZ domain secures the 3′ termini. The PIWI domain is characterized by its intrinsic endonuclease activity. Different AGO proteins exhibit preferences for the length and 5′-termini of the sRNA [89]. AGO1 and AGO10 prefer U sRNAs at the 5′-termini, AGO5 prefers C sRNAs at the 5’-termini, and AGO2/4/6/7/9 prefer A sRNAs at the 5’-termini. This base preference is related to the structure of nucleotide-specific recognition in the MID domain [90,91]. Mutants of *AGO1* in *Arabidopsis* display dwarf and sterile phenotypes [92]. Exclusive to the *AGO7* mutant is the manifestation of lobed leaf margins and an increased intercalary distance of lateral organs, in contrast to other *AGO* mutants which do not exhibit phenotypic deviations of significance when compared to the wild type [93]. AGO6 serves as a TGS suppressor with partial functional redundancy with AGO4 [94]. Rice and maize possess more AGO protein family members (19 and 17, respectively) than *Arabidopsis* [51]. In addition to proteins homologous to the three subfamilies in *Arabidopsis*, the grass family has a specific subfamily, *AGO18*. *ZmAGO18b* in maize, expressed in male ears, influences spikelet development [95]. Contemporary theoretical frameworks propose that *AGO10* safeguards *HD-ZIP-III-like* transcription factors from proteolytic breakdown by engaging in competitive binding with *AGO1* for the incorporation of *miR165/166* [76].

## 4. The Regulatory Function of sRNAs in Plant Development

### 4.1. Impact of sRNAs on Shoot Morphogenesis

The ontogeny of shoot development is typified by the orchestrated emergence of lateral organs, which are spatially and temporally patterned in distinct configurations at the shoot apical meristem (SAM). This developmental ballet necessitates the interplay of intricate and resilient genetic networks, which synchronize the proliferation programs across the diverse cellular and tissue landscapes. Within this intricate network, sRNAs stand out as key players (Figure 3). They constitute the foundational regulatory mechanisms that delineate shoot morphogenesis, conferring a degree of control and exactitude indispensable for the plant’s appropriate ontogeny and maturation. This specific role of sRNAs in regulating shoot development has elucidated the importance of these tiny molecules in the grand scheme of shoot morphogenesis [96].

The orchestration of shoot growth is contingent upon the meticulous organization and preservation of the SAM, a process quintessential to plant morphogenesis. The SAM, delineated during the early stages of embryogenesis, encompasses a reservoir of pluripotent stem cells that give rise to all aerial organs, ensconced within a milieu of undifferentiated meristematic cells [97]. It has been discovered that *PHABULOSA* (*PHB*) and *PHAVOLUTA* (*PHV*) are repressed by *miRNA165/166*, with *PHB* playing a significant role in SAM development [98,99,100]. AGO10 prevents *miRNA165/166* from binding to AGO1, thereby enabling its function. In *AGO10* mutants, increased activity of *miRNA165/166* precipitates a diminution in HD-ZIP III expression, concomitantly inhibiting the proliferation of the plant SAM [23,101]. Recent scholarly contributions have disclosed a contributory role for *miR394* in this regulatory cascade. The SAM is architecturally partitioned into three distinct strata (L1–L3). *miR394*, expressed in the outermost L1 layer, migrates to the L3 layer to repress its target, the F-Box protein LCR, and the transcription factor WUSCHEL (WUS) in the innermost L3 cells, thereby mediating stem cell functional diversity [102]. WUS enhances cellular proliferation within a reciprocal regulatory circuit with the *CLV* genes [103]. Phenotypic analyses of *miR394* mutants indicate that the repression of *LCR* by *miR394* is imperative for stem cell proliferation, underscoring that the sustenance of the SAM necessitates the synergistic interaction of both *WUS* and *miR394* [104,105]. Expressing the regulator *miR171* under the *WUS* promoter reduces the activity of the hairy meristem family, leading to premature termination of nutritive meristem organization. This implies that a consortium of microRNAs (miRNAs) may orchestrate cellular proliferation within the stem cell ecotone [106,107].

### 4.2. Functional Dynamics of sRNAs in Leaf Morphogenesis

Leaf morphogenesis, an exquisitely orchestrated process unfolding in a sequential cascade, entails phases of initiation, specification, transition, proliferation, and maturation. An array of coding genes, augmented by a curated cohort of non-coding sRNAs (Figure 3), has been implicated in this complex ballet of foliar development [5,108].

sRNAs and their intricate interplays not only dictate gene expression and regulatory frameworks but also assume pivotal roles in leaf morphogenesis via their synergistic integration with diverse genetic networks and physiological pathways [109]. *TAS3 siRNAs*, originating from miR390, are synthesized on the proximal aspect of juvenile foliage through the cleavage mediated by the AGO1/AGO7 complex. These siRNAs function to inhibit the transcription of factors ARF3/4, which are pivotal in fostering cellular polarity along the distal axis [110]. The interaction of *SPL* proteins regulated by *miR156* with CIN-*TCP* prevents CIN-*TCP* from interacting with *NAC* proteins to produce leaf serrations [111]. *miR159* promotes leaf cell proliferation by suppressing *MYB33* and *MYB65* expression. *miR160* selectively modulates members of the *ARF* family, specifically *ARF10/16/17*, thereby influencing leaf morphogenesis through the modulation of auxin signaling [112]. Concurrently, *miR164* orchestrates the expression of *NAC* domain genes, facilitating the delineation of leaf margins in *Arabidopsis* by targeting the *CUC* genes *CUC1*/*2*, which are integral *NAC* transcription factors implicated in the embryonic formation of the SAM and demarcation of developmental boundaries [113]. *miR165/166* represses HD-ZIP III family members on the distal side, causing proximal axialization of vascular bundles [101]. *miR319* is expressed as a *TCP* transcription factor and targets *CINCINNATA* (*CIN*)-*TCP* to promote cell differentiation, mitotic arrest, and to control leaf shape in angiosperms [114]. CIN-*TCP* also promotes leaf senescence via the jasmonic acid pathway. *miR393*, through its interaction with the target loci *TIR1* and *AFB1/2/3*, exerts a regulatory influence on foliar morphology and dimensions by modulating hormonal pathways [115]. The equilibrium between *miR396* and *GRFs* is instrumental in dictating the cellular composition of leaves and the dimensionality of meristematic tissues [116]. Additionally, *miR824* and its associated gene *AGL16* play a critical role in the ontogeny of stomatal structures [23].

### 4.3. Regulatory Mechanisms of sRNAs in Flowering Development

Floral ontogenesis represents a critical event within the life cycle of higher plant taxa, characterized by a complex coordination of gene networks. An extensive array of these characterized sRNAs has been associated with the regulation of diverse biological frameworks (Figure 3), mechanisms, and signaling pathways, functioning in concert with developmental cues [4]. Nonetheless, the preponderance of sRNA research in the context of floral development has predominantly centered on miRNAs, with scant attention accorded to siRNAs.

miRNAs play a crucial role in the regulation of floral organ identity and the timing of flowering in *Arabidopsis*. A significant number of miRNAs, such as miR156/157, 159, 164, 165/166, 167, 169, 171, 172, 319, 390, 393, and 824, are involved in the intricate control mechanisms that govern flowering processes [117]. The modulation of *miR156* expression, specifically its reduction, coupled with an elevation in the abundance of *SPL* proteins, instigates flowering. This is achieved through the activation of *LFY*, *FUL*, and *AP1*, thereby transitioning plant growth from vegetative to reproductive phases [118,119]. The influence of *miR156* and *SPL* on the development of rice flowers has been documented [120]. During periods of short daylight, *miR159* governs the floral transition by suppressing genes associated with floral meristem organization. Its target, *MYB33*, enhances the transcription of *ABA INSENSITIVE 5* (*ABI5*), which functions proximal to *miR156* in the regulatory hierarchy, orchestrating the transition from vegetative growth to the reproductive phase [121,122]. *miR164* orchestrates organ boundary formation by targeting *CUC* genes, with overexpression resulting in sepal fusion and a reduction in petal number [123]. *miR165/166* oversees floral morphogenesis and meristem activity [99]. *miR167* specifically targets the *ARF6* and *ARF8* genes that are integral to the regulation of pistils and stamens. Additionally, *miR167* is instrumental in directing the morphogenesis of both male and female reproductive structures in *Arabidopsis* [124]. Concurrently, *miR169* plays a pivotal role in modulating stress-induced floral induction by repressing the *AtNF-YA* transcription factor, culminating in the downregulation of *FLC* expression. This suppression facilitates the activation of *FLC* downstream target genes, including *FT* and *LFY*, thereby promoting the floral transition [125]. *miR171* targets *LOM1*, which regulates *SPL* transcription factor activity and delays flowering [118]. The overexpression of *miR172* in Arabidopsis leads to the precocious onset of flowering and the perturbation of floral organ morphogenesis. *miR172* suppresses *APETALA2* (*AP2*) expression by activating *SPL*, thereby promoting flowering time and floral organ determination. During specific phases of the plant growth cycle, *miR156* and *miR172* exhibit interactive regulatory roles, as evidenced by miRNA-mediated modulation [126]. *miR156* exerts an inhibitory effect on the expression of the *SPL* gene family, whereas certain *SPL* gene members reciprocally augment *miR172* expression. Within foliar tissues, the *SPL9–miR172b/c* complex is responsible for the regulation of flowering time-associated genes through the modulation of *FT* expression. Conversely, in apical meristematic regions, the *SPL15–miR172d* complex facilitates the floral initiation by inducing the expression of *MADS*-box genes. The transcriptional activity of the *miR172* gene is subject to regulation by environmental cues, such as ambient temperature and photoperiodicity [127,128]. *miR319* exerts regulatory control over the *TCP* gene family, facilitating the induction of flowering through transcriptional activation. This is achieved by the binding of *miR319* to the promoter region of the *CONSTANS (CO)* gene, thereby influencing its expression [129,130]. *miR390*-tasiRNA3-*ARF4* regulates flowering through the *AP1*/*FUL* pathway [23]. The overexpression of *miR393* has been demonstrated to impede the translation of *AFB2* and *TIR1*, consequently influencing the temporal regulation of flowering [131,132]. *AGL16* protein and its negative regulator, *miR824*, are pivotal in controlling the timing of flowering in *Arabidopsis*. *AGL16* is capable of direct interaction with *SVP* and an indirect association with *FLC*, leading to the formation of a repressive complex that inhibits the expression of *FT*. *miR824* and *AGL16* are integral in the suppression of floral induction during extended photoperiods [119,133]. In *Brachypodium distachyon*, a species within the Pooideae subfamily, the species-specific *miR5200* targets *FT*, thereby modulating the timing of flowering and influencing the floral transition [134].

### 4.4. sRNA-Mediated Regulation Mechanisms of Root Development

Recent scholarly focus has been directed towards the sRNA-mediated regulatory pathways in root development within plant biology research [135]. A comprehensive body of scientific literature has highlighted the critical functions of numerous miRNAs and ta-siRNAs in various aspects of root development (Figure 3). These facets encompass root morphogenesis, vascular tissue patterning, lateral root (LR) formation and extension, as well as adventitious root development [136].

An abundance of empirical evidence highlights the significant regulatory influence of miRNAs on the development of plant roots. *miR156* in conjunction with its target *SPL* gene plays a crucial role in the modulation of root length [137]. The dynamism of root meristematic cells is directed by *miR159* via its target gene *MYB65* [138]. *miR160* and its associated target genes *ARF10/16* are pivotal in affecting root elongation, influencing cellular division and differentiation processes at the root apex [139]. Mutations in *miR163* or overexpression of *PXMT1* result in diminished primary root lengths, indicating that *miR163* and its target gene *PXMT1* are integral to the configuration of root architecture during the nascent stages of *Arabidopsis* seedlings development. This provides further evidence that *HY5*-*miR163*-*PXMT1*-mediated post-transcriptional regulation affects root development [136,140]. In *Arabidopsis*, *miR164* is known to target quintuple mRNAs encoding NAC domain proteins. Mutants of *miR164a* and *miR164b* exhibit upregulated expression of *NAC1*, which correlates with an increased production of lateral roots [141]. Plant *miR165/166* enhances cell division and meristematic tissue activity by negatively regulating HD-ZIPIIIs to promote root elongation [142]. It also fosters xylem differentiation by repressing *PHABULOSA* (*PHB*) expression. *miR165* regulates *Arabidopsis* root differentiation [143]. The transcription factors *SHR* and *SCR* of the *GRAS* family play pivotal roles in root tip meristem formation [23,98]. *miR169* targets the *NF-YA10* gene and its target gene *NF-YA2* through the ABA pathway to control root morphological structure. The suppression of *NF-YA2* regulation by *miR169* indirectly influences the development of lateral roots [144]. Furthermore, *miR171* orchestrates the elongation of primary roots by inhibiting *HAM* genes, thereby modulating the activity of the quiescent center and overall root growth. Its target genes *SCL6*-II and *SCL6*-III are also involved [107,145]. *miR319b* and its target gene *MYB33* are implicated in the ethylene regulation of CBP20 phosphorylation, leading to a reduction in root tip number and a significant alteration in root morphology [146]. The *miR390*-associated trans-acting short interfering RNAs (*TAS3-tasiRNAs*) and their target genes *ARF2/3/4* provide quantifiable positive or negative feedback mechanisms that selectively modulate the length of lateral roots, without affecting their number or density [147]. *miR393* negatively regulates *TIR1* and *AFB* (*AFB1*, *AFB2*, and *AFB3*), critical genes of the growth hormone signaling pathway that promote primary and lateral root elongation in *Arabidopsis* seedlings. Under nitrate treatment, *Arabidopsis miR393* expression levels are upregulated, and morphological changes in both primary and lateral roots are observed in *AFB3* mutants, indicating that *miR393*/*AFB3* is induced by nitrate to regulate root architecture. Enhanced expression of *TIR1* augments the responsiveness to auxin treatment, culminating in the suppression of primary root elongation and a concomitant increase in the density of lateral roots. *TIR1* promotes *miR393* expression through a feedback pathway [115]. *Arabidopsis miR396a* targets seven *GRF* and *bHLH74* genes. Overexpression of *miR396a* results in shorter roots, while overexpression of *bHLH74* promotes root elongation. *miR396* is also involved in root development through interaction with *GRF*. The overexpression of *miR396* is correlated with a reduction in the cell cycle rate at the root apex and an enlargement of the root apical meristem. This suggests a significant role for miR396 in root development through the modulation of its target gene, *GRF* [148]. The accumulated evidence conclusively demonstrates that a vast array of miRNAs is essential in the regulation of plant root elongation and morphogenesis. These miRNAs function through their target genes, responding adaptively to the multifaceted environmental conditions within the root ecosystem.

Specific siRNAs have been elucidated as pivotal regulators in the modulation of root growth dynamics. For instance, in *Arabidopsis*, the *dcl2/3/4* mutant markedly diminishes the accumulation of siRNAs that influence RdDM, leading to an increase in cellulose and callus accumulation in the root xylem, thereby impacting root growth [149]. Particular siRNAs are capable of orchestrating plant root growth and development through the modulation of endogenous hormonal concentrations, including jasmonic acid, gibberellin, and ethylene [150]. In *Arabidopsis*, certain siRNAs are associated with root development and can influence root development by regulating various pathways, including root hair formation and cell division.

### 4.5. sRNA-Mediated Regulatory Mechanisms in Nodule Morphogenesis

Given the considerable energetic demands, symbiotic nitrogen fixation in legume nodules requires substantial carbon inputs, predominantly sourced from sugars. Legumes utilize the Autoregulation of Nodulation (AON) pathway, which is centrally modulated by miRNAs, to preserve a balanced nodule count [151,152]. These miRNAs are also instrumental in the morphogenesis and development of nodules (Figure 3). The emergence of high-throughput small RNA sequencing (sRNA-seq) technologies has significantly accelerated the identification of miRNAs associated with nodule formation [153].

The majority of research on miRNA regulation in the symbiosis between soybeans and rhizobia has been centered on nodulation. A host of miRNAs, including *miR160*, *167*, *171*, *172*, *393*, and *2111*, have been implicated in the developmental processes of rhizobia [153]. The *miR156b–SPL9d* complex, along with *NINa*, acts as a principal regulatory axis in the autoregulation of nodulation in soybeans [154]. Additionally, *miR172c* is known to inhibit the expression of *NNC1*, thereby facilitating the activation of the AON pathway [155]. *MiR2111* suppresses the negative regulator of nodulation, Too Much Love (*TML*), thereby maintaining the default susceptible state of legumes. Suppression of *miR2111* expression precipitates the accumulation of *TML* proteins in roots, concomitantly inhibiting the organogenesis of root nodules [156,157]. *MiR393j-3p* exerts an influence on nodulation by modulating the expression of the early nodulin gene *ENOD93* [158]. Conversely, the overexpression of *miR482*, *miR1512*, and *miR1515* is correlated with an increase in the number of nodules [159]. However, our understanding of how miRNAs, such as those involved in symbiotic nitrogen fixation (SNF) in mature nodules, regulate nodule functions remains limited. Future studies will elucidate the molecular mechanisms of miRNAs in root nodule formation and development and provide new strategies for regulating root nodule formation and development using miRNAs. In summary, the regulatory oversight of both the antecedent and subsequent elements within the AON pathway is attributed to the governance of sRNAs.

## 5. Conclusions and Future Perspectives

Recent accelerated progress in next-generation sequencing technologies has catalyzed significant breakthroughs in sRNA research within the plant sciences, particularly in crop species. Notably, many of these crops, such as oilseed rape and soybean, present challenges for genetic analysis owing to their protracted growth periods and intricate genomic structures. It is evident that agricultural crops exhibit both conserved and unique miRNA and siRNA pathways, analogous to those observed in other model plant species. These conserved pathways are integral to the regulation of vital growth and developmental processes, including, but not limited to, floral induction, root architecture formation, and pathogen resistance mechanisms. In contrast, lineage-specific or species-specific miRNAs and siRNAs hold significant biological interest due to their potential to bestow or modulate phenotypic characteristics that are absent or undeveloped in other plant taxa. Consequently, a comprehensive understanding of the regulatory dynamics between specific miRNAs/siRNAs and the manifestation of distinct phenotypic traits is imperative for the advancement of targeted breeding strategies.

Despite the considerable progress achieved in sRNA research within certain crop species, this progress remains disproportionately distributed, with a vast array of other crops yet to be thoroughly investigated. This disparity underscores the imperative for systematic sequencing and profiling of sRNAs, leveraging computational methodologies to discover new miRNAs and elucidate their intricate regulatory networks. It is paramount that these newly identified miRNAs undergo empirical validation within their respective host organisms, employing techniques, such as RNA interference (RNAi) or clustered regularly interspaced short palindromic repeats (CRISPR) for gene silencing, and various overexpression methods for gene activation. However, the execution of such analyses poses a significant challenge across numerous plant species, attributed to the necessity for robust translation mechanisms. To summarize, the preceding decade has witnessed only the nascent stages of sRNA exploration in agricultural crops. Prospective research endeavors must integrate comprehensive bioinformatic analysis and profound functional interpretation to fully demystify the roles and impacts of sRNAs.

## Figures and Tables

**Figure 1 ijms-25-07680-f001:**
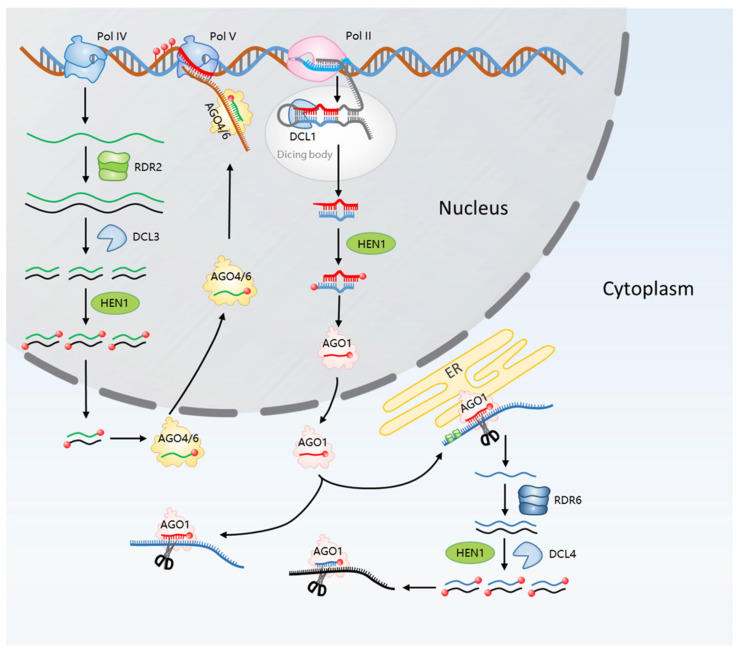
In plant cells, the biogenesis of miRNA and siRNA and their regulatory mechanisms are of paramount importance. The transcription of miRNAs is orchestrated by RNA polymerase II within the confines of the nucleus, culminating in the genesis of primary miRNAs (pri-miRNA). These pri-miRNAs are characterized by their distinctive hairpin structure. The transformation of double-stranded RNA into siRNA is expedited by the concerted efforts of DCL3, HEN, and RDR2. In the subsequent stages, the RISC–AGO complex assumes a pivotal role in guiding the specific strands of the siRNA duplex towards either post-transcription gene silencing (PTGS) or transcriptional gene silencing (TGS).

**Figure 2 ijms-25-07680-f002:**
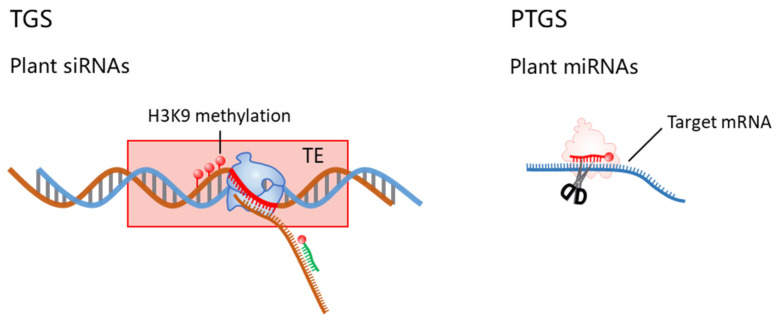
Small RNAs guide their effector proteins to specific loci to initiate either post-transcriptional gene silencing (PTGS) through transcript cleavage, RNA decay (degradation) and translation repression, or transcriptional gene silencing (TGS), which involves histone H3 Lys9 (H3K9) methylation at DNA loci that are homologous to the sRNA, often transposable element (TE) loci. Different mechanisms observed in different organisms are summarized in the same scheme.

**Figure 3 ijms-25-07680-f003:**
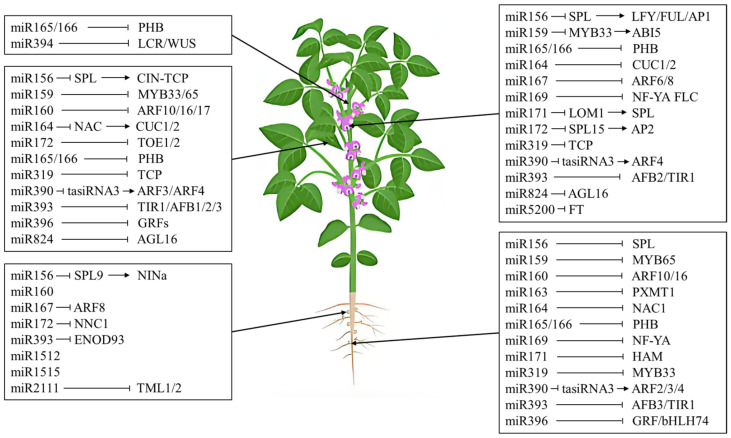
Functions of sRNAs in plant development and an overview of the current understanding of miRNA-mediated regulation in plants.

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
