# Peer review of "The Biosynthesis Process of Small RNA and Its Pivotal Roles in Plant Development"

_ijms, 2024, doi:10.3390/ijms25147680_

Round 1
Reviewer 1 Report
Comments and Suggestions for Authors
The review article titled “The biosynthesis process of small RNA and pivotal roles in plant development” reports an interesting work and focused on sRNA biosynthesis and its role in plant development. This is a well-written article and I anticipate that the manuscript should be of great interest to the researchers working on plant biotechnology and molecular biology. I include my comments below, most of them are suggestion to improve the overall quality for publication. I considered the manuscript suitable for publication subject to following improvements.
Title:
Suggestion: add “its” in the existing title or add another title from the suggested list
A. The biosynthesis process of small RNA and its pivotal roles in plant development
B. small RNA biosynthesis and its pivotal functions in plant development
C. Biosynthesis of small RNA and its impact on plant development
1. The authors elaborated abstract in a good way. However, revise the concluding remarks at the end of abstract section.
2. It is suggested to revise the statement Line 23-25 “This review endeavors to distill the contemporary comprehension of sRNA biosynthesis and underscore the pivotal role these molecules play in directing the intricate concert of plant developmental processes”.
3. Remove the word “distill” and add “evaluate”
4. Replace the word “concert” with “performance”
5. Rearrange the keywords alphabetically.
6. Revise the statement and cite appropriately, Line 30-31: “sRNAs, encompassing molecules of 20-30 nucleotides, have been recognized as fundamental modulators of gene expression in the plant kingdom”.
7. Line 37-38, revise the statement and replace the word “revelation” with discovery; “This revelation has significantly propelled the field of agricultural biotechnology forward, offering novel insights and applications in the modulation and enhancement of plant traits”.
8. It is suggested to revise the second paragraph of introduction section, and add information regarding miRNAs and siRNA. Moreover, delete the word odyssey and added recent literature in line with the study.
9. The objectives of the study should be added at the end of introduction section, revise it, to be readers friendly.
10. Add a cartoon figure for section 2.1 and 2.2 that shows difference or similarities in fuction or targets for miRNA and siRNA.
11. If you could add cartoon figures for key protein section 3 or add a table for describing their characteristics.
12. Overall, the manuscript is well designed and presented in a good way. However, many sentences include repetitive words and not explained and cited appropriately.
13. The article should become acceptable after English editing and minor revisions.
Author Response
Thank you very much for taking the time to review this manuscript. The point to point responds to your comments are listed as following:
Suggestion: add “its” in the existing title or add another title from the suggested list
Response: According to the reviewer’ s comment, we have added “its” in the existing title.
Comment 1-4. The authors elaborated abstract in a good way. However, revise the concluding remarks at the end of abstract section.
Response: Thank you for your valuable advice. We agree with comment 2 and have changed the word ‘distill’ to ‘evaluate’ as comment 3. Additionally, we replaced the word ‘concert’ with ‘performance’ as comment 4.
Comment 5: Rearrange the keywords alphabetically.
Response: Thank you for your advice. We have rearranged the keywords alphabetically.
Comment 6. Revise the statement and cite appropriately, Line 30-31: “sRNAs, encompassing molecules of 20-30 nucleotides, have been recognized as fundamental modulators of gene expression in the plant kingdom”.
Response: Thank you for your advice. We have revise the statement to 'sRNAs are found across all domains of life, including bacteria, archaea, and various eukaryotes. Their diverse compositions and functions have continued to astonish researchers over the past two decades.'
Comment 7. Line 37-38, revise the statement and replace the word “revelation” with discovery; “This revelation has significantly propelled the field of agricultural biotechnology forward, offering novel insights and applications in the modulation and enhancement of plant traits”.
Response: Thank you for your advice. We have replaced the word 'revelation' with 'discovery'.
Comment 8. It is suggested to revise the second paragraph of introduction section, and add information regarding miRNAs and siRNA. Moreover, delete the word odyssey and added recent literature in line with the study.
Response: Thank you for your advice. We have added information regarding miRNAs and siRNA in the second paragraph of introduction section. Furthermore, we have removed the word ‘odyssey’ and added references.
Comment 9. The objectives of the study should be added at the end of introduction section, revise it, to be readers friendly.
Response: Thank you for your advice. We have added the objectives of the study to the end of the introduction section.
Comment 10. Add a cartoon figure for section 2.1 and 2.2 that shows difference or similarities in fuction or targets for miRNA and siRNA.
Response: Thank you for your advice. We have added a cartoon figure for section 2.1 and 2.2 that shows difference in fuction or targets for miRNA and siRNA.
Comment 11. If you could add cartoon figures for key protein section 3 or add a table for describing their characteristics.
Response: Thank you for your advice. We’ve drawn cartoon diagrams of the key protein in Figure 1.
Comment 12-13. Overall, the manuscript is well designed and presented in a good way. However, many sentences include repetitive words and not explained and cited appropriately.
Response: Thank you for your positive comments. We have double-checked our sentences and references.
Reviewer 2 Report
Comments and Suggestions for Authors
The manuscript entitled "The biosynthesis process of small RNA and pivotal roles in plant development" is novel and hot-topic and also it provided comprehensive data about siRNAs and how they modulate plant development pathways. I think this manuscript is suitable for publication in its current form but to raise its attractivity, I highly recommend adding some data about the relation between siRNAs and lncRNAs. You might use the following published article as a paradigm; "From Trash to Luxury: The Potential Role of Plant lncRNA in DNA Methylation During Abiotic Stress".
Comments on the Quality of English LanguageThe English language is a bit hard to read.
Author Response
Thank you very much for taking the time to review this manuscript.
Comments:The manuscript entitled "The biosynthesis process of small RNA and pivotal roles in plant development" is novel and hot-topic and also it provided comprehensive data about siRNAs and how they modulate plant development pathways. I think this manuscript is suitable for publication in its current form but to raise its attractivity, I highly recommend adding some data about the relation between siRNAs and lncRNAs. You might use the following published article as a paradigm; "From Trash to Luxury: The Potential Role of Plant lncRNA in DNA Methylation During Abiotic Stress".
Response: Thank you for your positive comments. We have added some data on the relationship between siRNAs and lncRNAs at the end of section 2.2, citing the published article ‘From Trash to Luxury: The Potential Role of Plant lncRNA in DNA Methylation During Abiotic Stress’.